# Stratifying Risk for Pancreatic Cancer by Multiplexed Blood Test

**DOI:** 10.3390/cancers15112983

**Published:** 2023-05-30

**Authors:** Luca Digiacomo, Erica Quagliarini, Daniela Pozzi, Roberto Coppola, Giulio Caracciolo, Damiano Caputo

**Affiliations:** 1NanoDelivery Lab, Department of Molecular Medicine, Sapienza University of Rome, Viale Regina Elena, 291, 00161 Rome, Italy; luca.digiacomo@uniroma1.it (L.D.); erica.quagliarini@uniroma1.it (E.Q.); daniela.pozzi@uniroma1.it (D.P.); 2Fondazione Policlinico Universitario Campus Bio-Medico, Via Alvaro del Portillo, 200, 00128 Rome, Italy; r.coppola@policlinicocampus.it; 3Research Unit of Surgery, Department of Medicine and Surgery, Università Campus Bio-Medico di Roma, Via Alvaro del Portillo, 21, 00128 Rome, Italy

**Keywords:** pancreatic ductal adenocarcinoma, nanotechnology, early cancer diagnosis, biomarker, screening

## Abstract

**Simple Summary:**

Pancreatic ductal adenocarcinoma (PDAC) is a highly lethal disease, and the currently available techniques for its detection are either invasive or less sensitive. To overcome this limitation, we present here a multiplexed point-of-care test that combines systemic inflammatory response biomarkers, standard laboratory tests, and nanoparticle-based blood tests. This test provides a “*risk score*” for each individual under investigation, allowing clinicians to distinguish between PDAC patients and healthy subjects accurately and to determine the optimal diagnostic and therapeutic care pathway for each patient. As a result, this work may help advance progress in the early detection of PDAC and contribute to the development of screening programs for high-risk populations.

**Abstract:**

Pancreatic ductal adenocarcinoma (PDAC) is a highly lethal disease, for which mortality closely parallels incidence. So far, the available techniques for PDAC detection are either too invasive or not sensitive enough. To overcome this limitation, here we present a multiplexed point-of-care test that provides a “*risk score*” for each subject under investigation, by combining systemic inflammatory response biomarkers, standard laboratory tests, and the most recent nanoparticle-enabled blood (NEB) tests. The former parameters are routinely evaluated in clinical practice, whereas NEB tests have been recently proven as promising tools to assist in PDAC diagnosis. Our results revealed that PDAC patients and healthy subjects can be distinguished accurately (i.e., 88.9% specificity, 93.6% sensitivity) by the presented multiplexed point-of-care test, in a quick, non-invasive, and highly cost-efficient way. Furthermore, the test allows for the definition of a “*risk threshold*”, which can help clinicians to trace the optimal diagnostic and therapeutic care pathway for each patient. For these reasons, we envision that this work may accelerate progress in the early detection of PDAC and contribute to the design of screening programs for high-risk populations.

## 1. Introduction

Despite its relatively low incidence (i.e., 12.9 per 100,000 person-year), pancreatic cancer is among the leading cause of cancer-related deaths in both sexes worldwide [1,2] and exhibits the lowest five-year relative survival rate among all tumors [3,4]. The extremely poor prognosis of pancreatic cancer is largely related to both its aggressive behavior (local invasion, distant metastases) and the current difficulties in detecting it at an early stage [5,6]. Thus, a screening program for the early diagnosis of pancreatic cancer is urgently needed [6]. Manifold risk factors for pancreatic cancer have been identified, such as a family history [7], smoking [8], chronic pancreatitis [9], and diabetes mellitus [10], but currently, there are only a few recommended modalities for the screening of subjects at higher risk, such as diabetics or patients with specific genetic syndromes [11,12,13]. They generally include imaging methods, such as endoscopic ultrasonography and magnetic resonance cholangiopancreatography [13]. Furthermore, although developing strategies for early detection is considered critical for improving the survival of patients [2], the available techniques are not accurate enough. In this regard, carbohydrate antigen 19-9 (Ca 19.9) is the only marker approved by the Food and Drug Administration (FDA) for the management of pancreatic cancer [14,15]. However, it exhibits relatively low values of specificity and sensitivity [14], thus being more useful for patient follow-up than for screening and diagnosis [16]. To improve the diagnostic ability of CA 19.9, several biomarkers have been proposed [17], including altered/mutant genes, RNAs [18], proteins, lipids [19], and small metabolites [20,21]. Unfortunately, they still have not found real applicability in routine practice as they do not fulfill the World Health Organization (WHO) criteria for cancer screening and detection [22]. Those criteria require REASSURED techniques (i.e., Affordable, Sensitive, Specific, User-friendly, Rapid and robust, Equipment-free, and Deliverable to end-users techniques) [23].

On the other hand, nanotechnology has recently provided promising insights to develop non-invasive and cost-efficient tools for the detection of pancreatic cancers, especially pancreatic ductal adenocarcinoma (PDAC), the most prevalent neoplastic disease of the pancreas [24].

Among the most innovative methods, nanoparticle-enabled blood (NEB) tests have the potential to detect even small differences in the expression levels of circulating proteins that are associated with PDAC. NEB tests are based on differential protein adsorption on the surface of nanomaterials upon exposure to biological media, such as human plasma (HP). Indeed, once embedded in a biological medium, nanoparticles (NPs) act as accumulators of proteins for which they have a distinctive affinity and a low dissociation rate [25]. The protein layer that consequently forms on NPs is commonly referred to as protein corona and its composition has been demonstrated to contain information about the health status of individual subjects [26]. In other words, the protein corona is personalized [27]. This led to the intriguing possibility to exploit the protein corona technology to identify novel biomarkers for cancers by liquid chromatography (LC) in tandem with mass spectrometry (MS/MS) [28]. However, the scalability of biomarker discovery through LC-MS/MS raises significant sustainability concerns [22]. Interestingly, beyond the possible identification of biomarkers for cancers, the protein corona technology has been employed to classify non-oncological patients (NOP) from PDAC samples by evaluating the global protein patterns of the coronas with more affordable techniques, e.g., 1D sodium dodecyl sulphate–polyacrylamide gel electrophoresis (1D SDS-PAGE) [29,30]. The outcomes of NEB tests can be further paired to clinically relevant parameters in multiplexed strategies [22,31] to improve the classification ability of the techniques. In this work, we implemented a multiplexed test by using a gold NEB test [32,33] and common clinical parameters, including albumin, hemoglobin, glucose levels, and systemic inflammatory response biomarkers that have been demonstrated to have an important role in pancreatic cancer but are not enough specific to be used alone [34]. The choice of gold NPs relies on their ability to enhance the differences between plasma protein levels specifically attributed to PDAC, making them detectable by the protein corona technology. In this regard, it should be noted that the protein corona composition is affected by manifold factors, including the intrinsic properties of NPs (e.g., material, surface charge, functionalization, size) and environmental conditions (e.g., plasma concentration, temperature, pH, incubation time). In the past few years, our group has been working on optimizing these factors and demonstrated that gold NPs 100 nm in size are promising candidates as a nanoplatform for PDAC detection [32,33]. In this work, we show that implementing a gold NEB test into a multiplexed approach allowed for the classification between NOP and PDAC groups with improved specificity and sensitivity values (up to 89% and 94%, respectively). Furthermore, and more importantly, this approach can be employed to evaluate a “*risk score*” that may help clinicians design personalized diagnostic and therapeutic pathways for their patients.

## 2. Materials and Methods

### 2.1. Patients’ Enrollment and Inclusion Criteria

Cyto-histologically proven PDAC patients admitted to the Fondazione Policlinico Universitario Campus Bio-Medico that met the inclusion criteria reported in [35] have been considered eligible for the analysis. Their clinical and laboratory tests data were collected. The study was conducted according to the guidelines of the Declaration of Helsinki and approved by the Institutional Ethic Committee (prot. Prot.10/12ComEtCBM and further amendments).

### 2.2. Gold Nanoparticle-Enabled Blood Test

The gold NEB test of this study was previously developed and patented [32,33]. All the details of the experimental and analytical procedures can be found in these references. Briefly, gold NPs of size 100 nm (from Sigma-Aldrich (St. Louis, MI, USA), product ID 742031) were exposed to HP from PDAC and NOP donors for 1 h at 37 °C. Then, the protein corona was isolated by centrifugation, followed by three washing steps. The obtained pellets were re-suspended in loading buffer, boiled, and loaded on a gradient polyacrylamide stain-free gel for 1D SDS-PAGE analysis. Gel images were acquired by a ChemiDoc™ gel imaging system (Bio-Rad, CA, USA) and processed by means of custom MATLAB scripts (MathWorks, Natick, MA, USA) [36]. Finally, the electrophoretic readout was obtained as the integral areas of the protein corona patterns within specific ranges of molecular weight. 

### 2.3. Discriminant Analysis

Discriminant analysis was performed by the MATLAB (MathWorks, Natick, MA, USA) function “classify”, type “linear”, and prior probability “empirical”. All 17 clinical and electrophoretic parameters that are listed in Table 1 were inserted as input variables for 27 NOP and 47 PDAC subjects, whereas posterior probability and binary classification were obtained as outputs, then employed to compute ROC curves and risk scores.

## 3. Results

In total, 27 NOP and 47 PDAC patients were included in the study. A list of demographics and clinical characteristics for the study participants is reported in Table 2. Among the available laboratory test data, we selected the outcomes of standard tests (e.g., hemoglobin and glucose levels) and systemic inflammatory response biomarkers (e.g., white blood cells and neutrophils to lymphocytes ratio), as their relationship with pancreatic cancer has been widely investigated. Finally, four additional variables were included as predictors in the proposed multiplexed approach, i.e., the outcomes of an NEB test that has been specifically developed for the detection of PDAC. All the details of this technology are discussed in previous works [32,33], and can be summarized as follows. (i) Gold NPS (size = 100 nm) were incubated with HP from NOP and PDAC donors, (ii) the protein patterns that subsequently formed on their surface were evaluated by 1D SDS-PAGE, and (iii) information about the clinical status of the subjects was obtained by the upregulation or downregulation of corona proteins in specific molecular weight ranges. Our previous investigations revealed that the optimal electrophoretic outcomes are the integral areas of the molecular weight distributions within 10–20 kDa, 20–25 kDa, 25–35 kDa, and 35–45 kDa, and these four parameters are hereafter referred to as NEB_1_, NEB_2_, NEB_3_, and NEB_4_, respectively. A complete list of the selected clinical and electrophoretic parameters for NOP and PDAC samples is reported in Table 2, along with the corresponding *p*-values by Student’s *t*-test.

A graphical representation of those parameters is provided in Figure 1a, as a radial boxplot. Each radius represents a variable, whose minimum-to-maximum range is shown as a solid line and the 1st-to-3rd quartile as a filled box. Although statistically significant differences between NOP and PDAC were detected for some of the clinical (e.g., hemoglobin, and glucose level) and electrophoretic (e.g., NEB_1_) parameters, none of them alone classified the subjects with satisfactory values of specificity, sensitivity, and global accuracy. Indeed, receiver operating curves (ROC) corresponding to single-variable-based classifications (Figure 1b) exhibited small values of area under the curve (AUC). The best predictors were found to be NEB_1_ (AUC = 0.848), the blood levels of glucose (AUC = 0.833), and hemoglobin (AUC 0.794), but the corresponding classification accuracy was always below 80%. However, by considering simultaneously all the clinical and electrophoretic variables and performing a linear discriminant analysis, the classification ability of the test remarkably improved. In detail, only three NOP and three PDAC subjects were misclassified (Figure 2a), leading to specificity = 88.9% and sensitivity = 93.6%, with a consequent accuracy value that reached 91.9%. Accordingly, the AUC from ROC analysis read 0.980 (Figure 2b). Interestingly, the classification algorithm also provided a posterior probability value for each subject under study. That value falls between 0 and 1 and represents the probability of belonging to one of the two possible classes, i.e., NOP or PDAC. Thus, we propose the posterior probability value by discriminant analysis as a *risk score* for each subject under study. The *risk score* for all the participants is reported in ascending order (from left to right) in Figure 2c. Each dot corresponds to a patient, where green stands for NOP and red for PDAC subjects. In total, 22 out of 27 NOP (81.5%) exhibited a *risk score* below 0.3, whereas all the PDAC subjects overcame that risk value, and 39 out of 47 PDAC donors (83.0%) were found to have a *risk score* above 0.8. Furthermore, only one NOP subject (3.7%) had a *risk score* > 0.8. For these reasons, we defined a threshold that may be used to distinguish high-risk subjects from moderate- or low-risk subjects. Based on the outcomes of our analysis, we set the *risk threshold* to 0.8. Individuals who receive a screening score of 0.8 or higher are classified as screening positive, whereas those with a *risk score* lower than 0.8 are classified as screening negative.

## 4. Discussion

The early detection of pancreatic ductal adenocarcinoma (PDAC) remains a considerable challenge for the scientific community and is of utmost importance in clinical practice. Patients with PDAC usually access clinical consultation when the disease is already at an advanced stage, which restricts surgery to a small percentage. This emphasizes the critical need for the development of new screening programs that can identify PDAC at an early stage when symptoms are still mild. While certain clinical parameters, such as albumin, hemoglobin, glucose levels, and systemic inflammatory response biomarkers, have demonstrated their relevance in PDAC [37], they lack the necessary specificity to justify the use of invasive and costly second-level tests, such as US endoscopy and pancreatic MRI, for early detection. In recent years, nanotechnology has made significant strides in the advancement of novel diagnostic tools for the early detection of PDAC. Among them, nanoparticle-based blood (NEB) tests have emerged as promising methods for detecting PDAC in its early stages. NEB tests rely on the detection of subtle variations in circulating blood proteins by assessing their differential adsorption onto the surface of nanomaterials when exposed to biological fluids. The protein layer that forms, referred to as protein corona, has been demonstrated to contain information about the health status of individual subjects. In fact, it is well known that cancer produces alterations in the human proteome from the earliest stages. This led to the intriguing possibility of exploiting protein corona technology to identify novel biomarkers for cancers. In a typical NEB test, a nanoparticle is incubated with a clinically relevant biological fluid (e.g., human serum or plasma of healthy and oncological patients) at selected exposure conditions, including nanoparticle type, temperature, incubation time, etc., to form a protein corona that is further isolated and analyzed by simple proteomic techniques, such as 1D-SDS PAGE, to obtain different NEB readouts. 

In this study, we present the principles of a multiplexed strategy that combines the outcomes of a NEB test with clinically relevant parameters, thereby serving as a screening tool for PDAC. Our approach integrates the NEB test results based on the use of 100 nm gold nanoparticles with routine clinical analyses. Through our previous investigations, we have determined that the optimal indicators derived from a gold-based NEB test are the integral areas of four significant molecular weight distributions obtained from SDS PAGE profiles, namely 10–20 kDa, 20–25 kDa, 25–35 kDa, and 35–45 kDa, referred to as NEB_1_, NEB_2_, NEB_3_, and NEB_4_, respectively. Additionally, standard laboratory tests such as hemoglobin and glucose levels, as well as systemic inflammatory response biomarkers such as white blood cells and the neutrophil-to-lymphocyte ratio, were combined with the NEB readouts due to their established association with pancreatic cancer. Our results confirmed the power of the multiplexed strategy. In fact, even if statistically significant differences between healthy and PDAC subjects were detected by analyzing some of the clinical and electrophoretic parameters, none of them alone classified the subjects with suitable values of specificity, sensitivity, and global accuracy. However, by harnessing the power of the multiplexed test and employing discriminant classification analysis, we achieved high specificity, sensitivity, and AUC values of 88.9%, 93.6%, and 0.98, respectively.

These values outperformed specificity and sensitivity of CA 19.9, which have been reported equal to 77% and 76%, respectively [38]. We incidentally point out that the proposed multiplexed strategy did not include CA19.9 values, due to missing data for most of the subjects belonging to the non-oncological group. However, CA19.9 can be in principle inserted as a further input variable for future studies aiming at improving the technique. Nevertheless, in this form, the test allowed for the definition of a *risk score*, which varies between 0 and 1, and includes a threshold to distinguish high-risk subjects from moderate- or low-risk subjects. The threshold has been arbitrarily established considering the trend of the curve and to minimize the risk of false positives. Thus, it is possible that it could be modulated in the future after clinical trials on a larger scale that could lead to the identification of an optimal one. Due to the limited amount of early-stage subjects in the study, future research will also be aimed to enlarge the dataset and include more TNM I and TNM II PDAC donors. Despite this current limitation, the obtained *risk score* has a remarkable advantage, as it considers multiple clinical parameters, which are evaluated by standard laboratory analyses. Therefore, such risk stratification may be of great importance for clinicians looking for a first-level test to be used in selected high-risk subjects. 

## 5. Conclusions

In conclusion, we have presented the potential of a multiplexed test to be employed as a screening tool for PDAC. Our approach combines the readouts of routine clinical analyses with an NEB test based on the use of 100 nm gold NPs. The former includes the outcomes of standard laboratory tests and systemic inflammatory response biomarkers; the latter exploits the protein corona technology to detect even small changes in protein expression levels that are specifically due to PDAC. Globally, the employed discriminant classification analysis resulted in high specificity, sensitivity, and AUC values. Of note, the test enabled the determination of a *risk score* ranging from 0 to 1, providing clinicians with a valuable tool for risk stratification in high-risk populations (e.g., diabetics, patients with a familiar history of PDAC, etc.). The non-invasive, rapid, and cost-effective nature of this test underscores its potential significance as an initial screening tool. Thus, we can envision that subjects that resulted positively on the NEB test may be considered for second-level tests that cannot be considered screening tools due to their costs and invasiveness. Further investigations will be aimed at validating the technology on a large cohort and performing an accurate cost–benefit evaluation. These steps are fundamental to the design and application of an efficient screening program for PDAC in target populations.

## Figures and Tables

**Figure 1 cancers-15-02983-f001:**
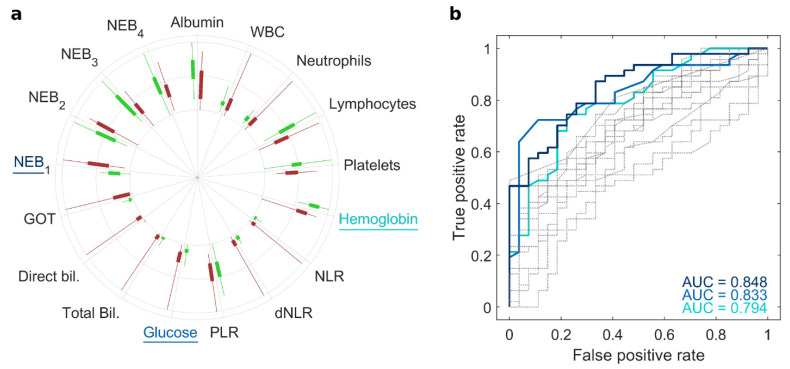
(**a**) Radial boxplot of clinical and electrophoretic parameters for NOP (green) and PDAC (red) subjects. (**b**) Receiver operating curves (ROC) obtained by classifying NOP and PDAC subjects according to each of the parameters. Parameters leading to the highest area under curve (AUC) are highlighted and correspond to hemoglobin (light blue), glucose level (blue), and NEB_1_ (dark blue).

**Figure 2 cancers-15-02983-f002:**
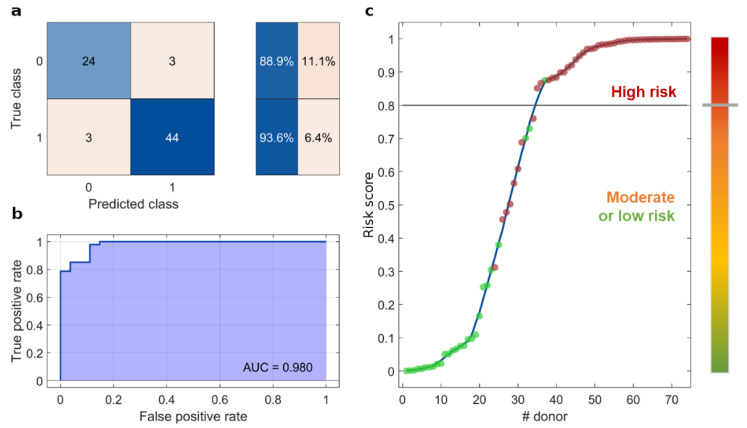
(**a**) Confusion matrix, (**b**) ROC curve, and (**c**) *risk score* obtained by the multiplexed test.

**Table 1 cancers-15-02983-t001:** List of mean values (±standard deviation) for the clinical and electrophoretic parameters that have been included in the classification analysis. *p*-values by Student’s *t*-test are reported.

	NOP (N = 27)	PDAC (N = 47)	*p*-Value
Albumin (g/mL)	3.62 ± 0.57	3.06 ± 0.65	0.0003
WBC (10^3^/μL)	7.28 ± 1.67	7.87 ± 3.54	0.4186
Neutrophils (10^3^/μL)	4.46 ± 1.23	5.51 ± 3.27	0.1134
Lymphocytes (10^3^/μL)	2.07 ± 0.64	1.64 ± 0.65	0.0075
Platelets (10^3^/μL)	253 ± 92.7	208 ± 68.9	0.0182
Hemoglobin (g/dL)	14.3 ± 1.76	11.9 ± 2.36	<0.0001
NLR	2.33 ± 0.95	4.12 ± 4.72	0.0564
dNLR	1.66 ± 0.55	2.61 ± 2.09	0.0235
PLR	134 ± 56.9	143 ± 62.7	0.5304
Glucose (mg/dL)	98.1 ± 12.9	131 ± 49.6	0.0011
Total bilirubin (mg/dL)	0.80 ± 0.62	2.31 ± 3.40	0.0249
Direct bilirubin (mg/dL)	0.16 ± 0.08	1.63 ± 2.85	0.0096
GOT (U/L)	19.5 ± 8.49	49.6 ± 56.5	0.0078
NEB_1_	0.22 ± 0.04	0.31 ± 0.08	<0.0001
NEB_2_	0.21 ± 0.05	0.21 ± 0.05	0.5040
NEB_3_	0.34 ± 0.09	0.28 ± 0.06	0.0007
NEB_4_	0.24 ± 0.08	0.20 ± 0.06	0.0231

**Table 2 cancers-15-02983-t002:** Demographics and clinical characteristics of the study participants.

	NOP (N = 27)	PDAC (N = 47)
Age, median (range)	58.5 (23–84)	71 (47–83)
Sex, N (%)		
Male	13 (48.1%)	23 (62.5%)
Female	14 (51.9%)	24 (37.5%)
Comorbidity, N (%)		
Cardiac	2 (7%)	15 (31.9%)
Pulmonary	1 (3.7%)	4 (8.5%)
Diabetes mellitus	1 (3.7%)	9 (19.1%)
Hypertension	3 (11.1%)	2 (4.2%)
None	15 (55.5%)	1 (2.1%)
Smoking status, N (%)		
Never	13 (48.2%)	12 (25.5%)
Ex-smoker	7 (25.9%)	16 (34%)
Current	7 (25.9%)	19 (40.5%)
CA 19.9 (UI/l), N (%)		
>37.00	0	36 (76.5%)
<37.00	5 (18.5%)	11 (23.5%)
Missing data	22 (81.5%)	0
TNM stage, N (%)		
I	NA	2 (4.4%)
II	NA	8 (17%)
III	NA	30 (63.8%)
IV	NA	7 (14.8%)

## Data Availability

Data will be available, according to the local policy, by contacting the corresponding authors.

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
