# Peer review of "Stratifying Risk for Pancreatic Cancer by Multiplexed Blood Test"

_cancers, 2023, doi:10.3390/cancers15112983_

Round 1
Reviewer 1 Report
Very interesting and important topic since PDAC is crucial problem of today and future medicine. The possibility of lab testing and ideally early detection of PDAC is a hot topic of several years, unfortunately until now without clear clinical benefit. Some research did not find really useful results, some technologies are so expensive and unreachable that it hampers their potential use in everyday practice. Therefore, presented method seems even more promissing and further testing is important.
Therefore, in genereal I recomment this work rof publication in high quality journal such as Cancers.
Several comments to make the paper of even more quality:
A) Improve introduction:
1. add some new citation about current screening programmes of PDAC, it is not entirely truth that there are none - e.g.
Biomedicines. 2022 Aug 23;10(9):2056. doi: 10.3390/biomedicines10092056.
2. some recent citation on use of miRNA profiles, to show that this is one of possible strategies - e.g.:
Biomedicines. 2021 Oct 14;9(10):1468. doi: 10.3390/biomedicines9101468.
3. some recent and quality citation on use of lipidemics, that seem very promising, however technology is challinging in many ways - e.g.:
Nat Commun. 2022 Jan 10;13(1):124. doi: 10.1038/s41467-021-27765-9.
4. cite some review of possible laboratory test strategies of early PDAC diagnosis - e.g.:
Can J Gastroenterol Hepatol. 2018 Aug 14;2018:5389820. doi: 10.1155/2018/5389820. eCollection 2018.
B) improve discussion
1. you must comment on the fact, that most of the PDAC patients were late stages, therefore, it is far from sure, that this test will help early PDAC diagnostics - list it as a logical weakness of the study that will be addressed in future research.
2. unfortunatelly you dont have CA19-9 in majority of controls, just from the few available it is logically low and in majority of patients it is high. How can we know that CA19-9 isnt doing the same quality discrimination of high stage PDAC patients and healthy controls as your novel test? You must address this, if you are not able to get CA19-9 from your healthy controls, the ask statistician if the CA19-9 numbers you have is enough for at least some statistics proving your test better than CA19-9, if this would show impossible then address it as a weakness of the study and compare at least with literature data on CA19-9 specificity and sensitivity.
Author Response
Reviewer 1
Very interesting and important topic since PDAC is crucial problem of today and future medicine. The possibility of lab testing and ideally early detection of PDAC is a hot topic of several years, unfortunately until now without clear clinical benefit. Some research did not find really useful results, some technologies are so expensive and unreachable that it hampers their potential use in everyday practice. Therefore, presented method seems even more promising and further testing is important.
We thank the reviewer for his/her comments.
Therefore, in general I recommend this work for publication in high quality journal such as Cancers. Several comments to make the paper of even more quality:
- A) Improve introduction:
- add some new citation about current screening programmes of PDAC, it is not entirely truth that there are none - e.g. Biomedicines. 2022 Aug 23;10(9):2056. doi: 10.3390/biomedicines10092056.
- some recent citation on use of miRNA profiles, to show that this is one of possible strategies - e.g.: Biomedicines. 2021 Oct 14;9(10):1468. doi: 10.3390/biomedicines9101468.
- some recent and quality citation on use of lipidomics, that seem very promising, however technology is challenging in many ways - e.g.: Nat Commun. 2022 Jan 10;13(1):124. doi: 10.1038/s41467-021-27765-9.
- cite some review of possible laboratory test strategies of early PDAC diagnosis - e.g.: Can J Gastroenterol Hepatol. 2018 Aug 14;2018:5389820. doi: 10.1155/2018/5389820. eCollection 2018.
A 1-4. We thank the reviewer for his/her suggestions. We added all the proposed citations in the revised manuscript.
- B) improve discussion
- you must comment on the fact, that most of the PDAC patients were late stages, therefore, it is far from sure, that this test will help early PDAC diagnostics - list it as a logical weakness of the study that will be addressed in future research.
B1. We gratefully thank the reviewer for his/her comment. It is right that most of the enrolled patients were late-stage PDAC subjects (i.e. TNM III, 63.8%). We specified this limitation in the discussion of the revised manuscript and point out that future research will involve the study on a large cohort that includes higher numbers of early-stage PDAC patients.
- Unfortunately you do not have CA19-9 in majority of controls, just from the few available it is logically low and in majority of patients it is high. How can we know that CA19-9 is not doing the same quality discrimination of high stage PDAC patients and healthy controls as your novel test? You must address this, if you are not able to get CA19-9 from your healthy controls, the ask statistician if the CA19-9 numbers you have is enough for at least some statistics proving your test better than CA19-9, if this would show impossible then address it as a weakness of the study and compare at least with literature data on CA19-9 specificity and sensitivity.
B2. We thank the reviewer for his/her comment. We agree with his/her considerations about the small number of available CA19-9 values for non-oncological patients. For 22 out of 27 non-oncological patients, data are missing. For this reason, CA19-9 cannot be included as an input variable in the multiplexed approach. Indeed, the proposed method requires that all the input parameters have to be available for all the subjects. Thus, inserting CA19-9 in the study would lead to a severe loss of sample size. This is a limitation of the work, and –as suggested- has been specified in the revised manuscript, along with a comparison with literature data on CA19-9.
Nonetheless, in our opinion the absence of data related to CA 19-9 should not affect the obtained results and can also be in agreement with the main aim of our investigation. In other words, considerinf that as previously reported, elevated CA19-9 is useful in following patients with known disease (Ryan, David P., Theodore S. Hong, and Nabeel Bardeesy. "Pancreatic adenocarcinoma." New England Journal of Medicine 371.11 (2014): 1039-1049.) and that its sensitivity and specificity alone in diagnosing PDAC have been reported to be low (i.e. about 78.2% and 82.8%, respectively) (Yang, Chi-Ying, et al. "Accuracy of simultaneous measurement of serum biomarkers: Carbohydrate antigen 19-9, pancreatic elastase-1, amylase, and lipase for diagnosing pancreatic ductal adenocarcinoma." Journal of the Formosan Medical Association (2022) its efficacy in in predicting either pancreatic cancer or other cancers in the asymptomatic population is low. This concept, even if old (Chang, Chi-Yang, et al. "Low efficacy of serum levels of CA 19-9 in prediction of malignant diseases in asymptomatic population in Taiwan." Hepato-gastroenterology. 2006) is still taken into account also by the WHO that do not consider CA 19-9 in the diagnostic setting, also because of there is a 10% of the population that does not secret this biomarker. Taken together, these considerations can suggest that the multiplexed test have better diagnostic potential with other biomarkers.

Reviewer 2 Report
The title does not appropriate to the content.
NEB was abbreviated but not appropriate.
What was the role of gold nanoparticles? Why particularly used that size?
The obtained results in the study are not enough to claim the content.
What is the novelty of this study?
Author Response
Reviewer 2
The title does not appropriate to the content.
We thank the reviewer for his/her suggestion. We modified the title. The revised version of the manuscript is entitled: “Stratifying risk for pancreatic cancer by multiplexed blood test”. It suggests the use of a multiplexed blood test to categorize or divide individuals into different risk groups based on their likelihood of developing pancreatic cancer. This approach aims to identify those at higher risk who may benefit from closer monitoring or early interventions, while potentially sparing those at lower risk from unnecessary procedures.
NEB was abbreviated but not appropriate.
We thank the reviewer for his/her comment. The expression that has been used in the abstract (nanoparticle-based test) did not match with the acronym, i.e. NEB (nanoparticle-enabled blood) test. The mistake has been corrected.
What was the role of gold nanoparticles? Why particularly used that size?
We thank the reviewer for his/her question. The proposed test involves the employment of the protein corona technology. The protein corona is a biomolecular layer that forms on nanoparticles (NPs) when they interact with biological fluids, e.g. human plasma. The composition of the protein corona depends on protein-nanoparticle affinity, it is personalized and contains information about the health status of single subjects (references are provided in the manuscript). In other words, NPs in plasma act as accumulators of proteins and thus can be used to detect even small differences in the protein expression levels that are specifically related to the disease.
However, it should be noted that the protein corona composition is affected by manifold factors, including the intrinsic properties of NPs (e.g. material, surface charge, functionalization, size) and environmental conditions (e.g. plasma concentration, temperature, pH, incubation time). Therefore, it is of fundamental importance to choose an optimal nanomaterial as “nano-accumulator” and to tune the experimental conditions of incubation. In the last few years, our group has been working on these aspects and found 100 nm-sized gold nanoparticles to be a promising platform for the development of NP-based blood tests. In this regard, some of the main results about the employment of gold NPs for the detection of PDAC and the corresponding effects of the experimental conditions can be found in the following publication and patent:
- Digiacomo, L.; Caputo, D.; Coppola, R.; Cascone, C.; Giulimondi, F.; Palchetti, S.; Pozzi, D.; Caracciolo, G. Efficient pancreatic cancer detection through personalized protein corona of gold nanoparticles. Biointerphases 2021, 16, 011010.
- Digiacomo, L.; Palchetti, S.; Giulimondi, F.; Pozzi, D.; Chiozzi, R.Z.; Capriotti, A.L.; Laganà, A.; Caracciolo, G. The biomolecular corona of gold nanoparticles in a controlled microfluidic environment. Lab on a Chip 2019, 19, 2557-2567
- Caracciolo, G.; Pozzi, D.; Palchetti, S.; Digiacomo, L.; Caputo, D.; Coppola, R. A method to assist in the early diagnosis of pancreatic adenocarcinoma. 2022
In conclusion, the choice of gold NPs relies on their ability to enhance the differences between plasma protein levels specifically attributed to PDAC, making them detectable. We apologize that this aspect was not clear in the submitted manuscript. We included the aforementioned aspects in the introduction of the revised article.
The obtained results in the study are not enough to claim the content.
We acknowledge the reviewer for his/her comment. We point out that results refer to a dataset of 17 parameters, which have been evaluated for each of the 74 donors included in the study. The details about patients’ enrollment, inclusion criteria, demographics, clinical characteristics, and statistical analysis are provided, along with the description and discussion of the obtained outcomes. In addition, in the revised version of the manuscript, we outlined some of the main limitations of the work. They include the relatively small sample size, the wide abundance of late-stage patients with respect to the early-stage ones, and the lack of CA19.9 data for most of the non oncological subjects. As specified, these aspects will be taken into account in the next studies. However, the presented data support the main conclusions of the work, i.e. (i) the employed discriminant classification analysis resulted in high specificity, sensitivity, and AUC values and (ii) the non-invasive, rapid, and cost-effective nature of this test underscores its potential significance as an initial screening tool. Furthermore, (iii) the determination of a risk score may provide clinicians with a valuable tool for risk stratification in high-risk populations (e.g., diabetics, patients with a familiar history of PDAC, etc.).
What is the novelty of this study?
We thank the reviewer for his/her question. The novelty of the study relies on the possible employment of a blood test that quantifies simultaneously manifold variables (including the electrophoretic readouts from a nanoparticle-enabled blood test) and provides a risk score for PDAC. The test would be non-invasive, affordable and robust. As better specified in the revised manuscript, the evaluation of a risk score can be useful to help clinician trace the optimal diagnostic (e.g. by second-level tests) and therapeutic pathways for each patient.

Reviewer 3 Report
The research work carried out on “Stratifying risk for pancreatic cancer by nanoparticle-based 2 multiplexed testing” is a significant cotribution. The study's strengths lie in the use of multiple biomarkers and tests, which have the potential to improve the accuracy of pancreatic ductal adenocarcinoma (PDAC) detection and aid in designing screening programs for high-risk populations. However, the study's limitations include a small sample size and the need for further validation. I have some minor comment on the manuscript:
1. There is significant divergence in the risk score of approximately 18.5% for NOP and PDAC. How author explain these findings. If this is the result of small sample size chosen by authors?
2. The discussion should have lucid language and should comprehensively reflect on the results.
Author Response
Reviewer 3
The research work carried out on “Stratifying risk for pancreatic cancer by nanoparticle-based 2 multiplexed testing” is a significant contribution. The study's strengths lie in the use of multiple biomarkers and tests, which have the potential to improve the accuracy of pancreatic ductal adenocarcinoma (PDAC) detection and aid in designing screening programs for high-risk populations. However, the study's limitations include a small sample size and the need for further validation. I have some minor comment on the manuscript:
We thank the reviewer for his/her comments. Study limitations, including the small sample size have been emphasized in the discussion and conclusion sections of the revised manuscript. Overcoming these limitations will be among the main aims of our next studies.
- There is significant divergence in the risk score of approximately 18.5% for NOP and PDAC. How author explain these findings. If this is the result of small sample size chosen by authors?
We thank the reviewer for his/her comment. According to the obtained outcomes, only 1 NOP subject exhibited a risk score above threshold and 8 PDAC subjects exhibited a risk score below threshold. Globally, 9 samples out of 74 (about 12%) had a risk score that did not reflect their actual class. However, the spread of the risk score distributions for NOP and PDAC was about 0.24 and 0.17, respectively. As suggested by the reviewer, increasing the sample size would improve these outcomes, and lower the divergence. The importance of enlarge the data set has been emphasized in the revised version of the manuscript.
- The discussion should have lucid language and should comprehensively reflect on the results.
We thank the reviewer for his/her comment. As suggested, we have revised the Discussion section to provide a clearer and more detailed reflection on the results.

Round 2
Reviewer 2 Report
No comments. Good Luck.